# Evaluation of Ouabain’s Tissue Distribution in C57/Black Mice Following Intraperitoneal Injection, Using Chromatography and Mass Spectrometry

**DOI:** 10.3390/ijms25084318

**Published:** 2024-04-13

**Authors:** Denis A. Abaimov, Rogneda B. Kazanskaya, Ruslan A. Ageldinov, Maxim S. Nesterov, Yulia A. Timoshina, Angelina I. Platova, Irina J. Aristova, Irina S. Vinogradskaia, Tatiana N. Fedorova, Anna B. Volnova, Raul R. Gainetdinov, Alexander V. Lopachev

**Affiliations:** 1Research Center of Neurology, Volokolamskoye Shosse 80, 125367 Moscow, Russia; abaimov@neurology.ru (D.A.A.); timoshina.yu.a@neurology.ru (Y.A.T.); fedorova@neurology.ru (T.N.F.); 2Biological Department, Saint Petersburg State University, Universitetskaya Emb. 7/9, 199034 St. Petersburg, Russia; aristovy@hotmail.com (I.J.A.); a.volnova@spbu.ru (A.B.V.); 3Scientific Center for Biomedical Technologies of the Federal Biomedical Agency of Russia, 119435 Krasnogorsk, Russia; ageldinov@gmail.com (R.A.A.); mdulya@gmail.com (M.S.N.); 4Biological Department, Lomonosov Moscow State University, Leninskiye Gory 1, 119991 Moscow, Russia; 5The Mental Health Research Center, Kashirskoye Shosse 34, 115522 Moscow, Russia; platova@psychiatry.ru; 6Institute of Translational Biomedicine, Saint Petersburg State University, Universitetskaya Emb. 7/9, 199034 St. Petersburg, Russia; gainetdinov.raul@gmail.com; 7Non-State Private Educational Institution of Higher Professional Education, Moscow University for Industry and Finance “Synergy”, Meshchanskaya Street, 9/14, Building 1, 129090 Moscow, Russia; irina_www@mail.ru; 8Saint-Petersburg University Hospital, 199034 St. Petersburg, Russia

**Keywords:** cardiotonic steroids, brain–blood barrier, ouabain, digoxin, brain, blood plasma, kidney, cardiac tissue, liver

## Abstract

Cardiotonic steroids (CTSs), such as digoxin, are used for heart failure treatment. However, digoxin permeates the brain–blood barrier (BBB), affecting central nervous system (CNS) functions. Finding a CTS that does not pass through the BBB would increase CTSs’ applicability in the clinic and decrease the risk of side effects on the CNS. This study aimed to investigate the tissue distribution of the CTS ouabain following intraperitoneal injection and whether ouabain passes through the BBB. After intraperitoneal injection (1.25 mg/kg), ouabain concentrations were measured at 5 min, 15 min, 30 min, 1 h, 3 h, 6 h, and 24 h using HPLC–MS in brain, heart, liver, and kidney tissues and blood plasma in C57/black mice. Ouabain was undetectable in the brain tissue. Plasma: C_max_ = 882.88 ± 21.82 ng/g; T_max_ = 0.08 ± 0.01 h; T_1/2_ = 0.15 ± 0.02 h; MRT = 0.26 ± 0.01. Cardiac tissue: C_max_ = 145.24 ± 44.03 ng/g (undetectable at 60 min); T_max_ = 0.08 ± 0.02 h; T_1/2_ = 0.23 ± 0.09 h; MRT = 0.38 ± 0.14 h. Kidney tissue: C_max_ = 1072.3 ± 260.8 ng/g; T_max_ = 0.35 ± 0.19 h; T_1/2_ = 1.32 ± 0.76 h; MRT = 1.41 ± 0.71 h. Liver tissue: C_max_ = 2558.0 ± 382.4 ng/g; T_max_ = 0.35 ± 0.13 h; T_1/2_ = 1.24 ± 0.7 h; MRT = 0.98 ± 0.33 h. Unlike digoxin, ouabain does not cross the BBB and is eliminated quicker from all the analyzed tissues, giving it a potential advantage over digoxin in systemic administration. However, the inability of ouabain to pass though the BBB necessitates intracerebral administration when used to investigate its effects on the CNS.

## 1. Introduction

Cardiotonic steroids (CTSs), commonly called cardiac glycosides, are a group of compounds used for treating heart failure. They are primarily found in plants, such as digitalis, although mammals also possess endogenous CTSs [1]. CTSs are composed of a steroidal core and a lactone ring and may include 1–3 monosaccharide residues. CTSs, as a whole, can be subdivided into two groups based on the number of carbon atoms in the lactone ring: cardenolides (5-carbon lactone ring) and bufadienolides (6-carbon lactone ring).

The pharmacological target of CTSs is the Na^+^/K^+^-ATPase (NKA) enzyme, which has a CTS-binding pocket in its extracellular section. CTS binding causes NKA inhibition and activates a number of intracellular signaling pathways, including MAPK [2,3]. In the heart, this can cause cardiac muscular hypertrophy [4,5]. 

However, although low concentrations of the CTS digoxin are used for treating heart failure [6], because of its positive ionotropic effect, the applicability of CTSs in the clinic is limited by their adverse side effects, including those on the central nervous system (CNS). Some of the CNS side effects include headaches, dizziness, brain fog, sleep disturbances, and delirium [7,8]. The prevalence of these side effects is mediated by the permeability of the blood–brain barrier (BBB) to most CTSs. 

Ouabain, like digoxin, is a cardenolide. However, unlike digoxin, ouabain has a CH_2_OH group at the C10 position, hydroxyl groups at C1 and C5, and rhamnose, increasing its polarity (Figure 1). Studies utilizing the direct intracerebroventricular administration of the CTS ouabain to rats and mice demonstrated that CTSs can cause changes in dopaminergic signaling, accompanied by mania-like behaviors [9,10]. As the BBB is known to be permeable primarily to lipophilic molecules, the increased polarity of ouabain compared to other CTSs suggests that it will not pass through the BBB as easily. This makes ouabain interesting as a potential alternative to digoxin in treating patients. Previously, ouabain has been applied in clinical practice in Germany under the trademark “Strodival” (Meda Pharma GmbH, Solna, Sweden). However, its use in Germany was discontinued in 2006. At the present moment, ouabain is clinically used only in Ukraine, where it is registered as “Strophantin-G”.

It is known that the pharmacokinetics of a given drug are significantly affected by the administration route. As such, the adsorption and excretion of ouabain when administered in different ways has been extensively investigated. Peroral and sublingual administration routes were shown to be ineffective, with ouabain’s enteral adsorption barely approaching 7% at 5 h post administration [11]. Absorption coefficients for peroral administration are reported to range from 0.7% to 3.0% (2.2% on average). A comparison of ouabain’s kidney excretions after intravenous and peroral administrations, using radioimmunoassays, showed that the intravenous route is superior, with kidney excretion post i.v. administration being 66% of the initial dose, while excretion post peroral administration was only 1.3% of the initial dose [12]. When ouabain is administered intravenously, its pharmacokinetics are best described using the two-compartment model. Ouabain plasma concentrations drop rapidly post injection, with its half-life in the alpha-phase reportedly ranging from 2 to 20 min. This makes evaluating the beta-phase half-life difficult, as ouabain concentrations during it are close to undetectable. As such, its half-life in the beta-phase varies from 11 to 50 h [13]. 

Most investigations conducted on the pharmacokinetics of ouabain, both in animals and humans, were performed in the time period 1970–1980 and used tritium-labeled ouabain and radioisotopic detection methods, which are unable to exclude potentially pharmacologically inactive metabolites from the total measured radiation. As mentioned above, ouabain is more polar than other CTSs, making it the most likely to be resistant to passing through the blood–brain barrier. As such, it is possible that the previous studies showing the penetration of ouabain in brain tissue obtained false positives. Contemporary pharmacological evaluations of compounds of interest frequently use chromatography and mass spectrometry, which are more specific and possess fewer limitations in comparison to radioisotopic detection. Thus, investigating the ability of ouabain to penetrate the BBB, its kinetics in blood plasma, and its distribution in organs, using HPLC–MS, became the goal of this study. 

This study investigated the distribution and elimination kinetics of ouabain in plasma as well as in heart, kidney, and liver tissues, using a direct HPLC–MS method rather than radioactive labeling as in earlier studies. The tropicity and bioavailability of ouabain with respect to various organs and tissues differ significantly from those of digoxin. Most importantly, we showed that ouabain does not pass through the BBB, meaning that it lacks CNS side effects when administered systemically. This, in conjunction with ouabain’s tropism in cardiac tissue and lack of cardiac muscular hypertrophy induction, suggests that ouabain may be preferable to digoxin in clinical applications.

## 2. Results

To facilitate accurate ouabain detection, we developed a custom high-efficiency liquid chromatography–mass spectrometry (HPLC–MS) protocol, as described in the Materials and Methods section. Ouabain concentrations were evaluated in the blood plasma and brain, heart, liver, and kidney tissues of C57/black mice at a range of time-points post intraperitoneal injection of 1.25 milligrams of ouabain per kilogram of bodyweight (5 min, 15 min, 30 min, 1 h, 3 h, 6 h, and 24 h).

In the blood plasma, the maximum concentration (C_max_) of the ouabain was 882.88 ± 21.82 ng/g (T_max_ = 5 min post injection and 0.08 ± 0.01 h). After peaking at 5 min, it decreased monoexponentially (Figure 2A), dropping to 14 ng/g at 1 h post injection. The half-life (T_1/2_) of the ouabain in the plasma was determined at 0.15 ± 0.02 h. The mean retention time (MRT) was 0.26 ± 0.01 h. In sum, this implies that within 1 h following the intraperitoneal injection, ouabain almost entirely ceases circulation in the blood plasma of animals (Figure 2A). Other pharmacokinetic parameters for ouabain elimination from blood plasma are provided in Table 1.

In the perfused brain tissue, ouabain was not detectable at any time points after the injection. As such, it can be concluded that ouabain does not pass through the BBB when administered systemically to mice and does not accumulate in the brain tissue.

In the cardiac tissue, which is the “target” CTS tissue, like in the blood plasma, T_max_ was 5 min post injection (0.08 ± 0.02 h), and C_max_ = 145.24 ± 44.03 ng/g. After 5 min, the concentration of the ouabain decreased steadily and became undetectable at 60 min (Figure 2B). T_1/2_ for the cardiac tissue was 0.23 ± 0.09 h, and MRT was 0.38 ± 0.14 h. Other pharmacokinetic parameters for ouabain elimination from the cardiac tissue are provided in Table 1. Thus, ouabain is present in animal cardiac tissue for the same amount of time following injection as in the blood plasma and does not appear to accumulate.

In the kidney tissue, C_max_ was 1072.3 ± 260.8 ng/g, and T_max_ was 0.35 ± 0.19 h (Figure 2C). T_1/2_ was 1.32 ± 0.76 h, and MRT was 1.41 ± 0.71 h. Other pharmacokinetic parameters for ouabain elimination from the kidney tissue are provided in Table 1. In the liver tissue, the C_max_ of the ouabain was 2558.0 ± 382.4 ng/g at 30 min post injection (T_max_ = 0.35 ± 0.13 h) (Figure 2D). T_1/2_ was 1.24 ± 0.7 h, and MRT was 0.98 ± 0.33 h. Other pharmacokinetic parameters for ouabain elimination from the liver tissue are provided in Table 1. In conclusion, ouabain accumulates in both kidney and liver tissues, which is expected as they are both organs of elimination. The greater retention time of the ouabain by the liver tissue in comparison to the blood plasma indicates that the liver is the main accumulation site (Figure 2C,D).

## 3. Discussion

In this study, we used a custom HPLC–MS protocol to evaluate ouabain elimination from C57/black mice tissues following intraperitoneal injection (1.25 mg/kg bodyweight). In this study, we express the concentration values in nanograms of ouabain per gram of tissue, which affects our interpretation of the volumetric distribution (Vd) and clearance (CL) values. although Vd is usually expressed in liters or liters per kilogram, and Cl is expressed in liters per hour or liters per hour per kilogram, in our study, Vd should be interpreted as the hypothetical mass of a test tissue in which the administered dose of the drug could be distributed to achieve the same concentration level in that test tissue. 

Ouabain’s Vd/F value for plasma was 900 ± 153 g/kg. That is, the hypothetical volume of the test tissue to achieve the concentration measured in that tissue when the same dose is administered would have to be about 900 grams per kilogram of bodyweight. In other words, this is the fraction of the volume that would be taken up by the administered substance to achieve the concentrations obtained if the entire body of the animal were represented only by the test tissue. We found that in the blood plasma, the pharmacokinetics of the ouabain are best described using a one-compartment model. After reaching its maximum during the first 5 min, the concentration of the ouabain decreases more than 10-fold within an hour, indicating that its circulation time in the plasma is relatively short. This assumption is supported by the short half-life (T_1/2_ = 0.15 ± 0.02 h) and mean retention time (MRT = 0.26 ± 0.01 h). This finding is consistent with previous studies performed using radioautographic methods, which indicate that ouabain is rapidly eliminated from the blood plasma within the first hour following systemic administration. In contrast, for digoxin, the MRT in the blood plasma is 10 times higher [14].

Digoxin is known to have side effects on the CNS in patients treated for heart failure [15]. According to Andersson et al., the concentration of digoxin in the brains of inpatients regularly taking digoxin can reach a level of 32 ng/g, which is comparable to that observed in skeletal muscle [16]. This elevation in the brain’s digoxin concentration causes neuropsychiatric side effects, such as fatigue, depression, psychosis, and delirium [2,17]. A key finding of this study is the apparent inability of ouabain to cross the BBB in mice. This contradicts data acquired using radioautographic methods in other studies, which found trace amounts (6–7 ng/g) of ouabain in brain tissue following systemic administration [18] and is significantly higher than the lower detection threshold of our protocol. It should be noted that no previous studies have been performed specifically on mice, and making a direct comparison may be inappropriate. Rodents, including mice and rats, have adapted to a CTS-containing diet, as reflected in their CTS-resistant α1 NKA subunit [19]. It is possible that their BBB is also less permeable to CTS; however, digoxin does pass through the BBB of mice [14]. In cats, it was shown that most cardiac glycosides are able to cross the blood–brain barrier, with an efficiency directly proportional to their lipophilicity [20], and ouabain is less lipophilic than digoxin. Ouabain detected in the brain in previous studies may be caused by lipophilic metabolites of the radioactive-labeled ouabain, such as ouabagenin, penetrating the BBB. The investigators also may have obtained false positives due to poor cerebral perfusion and residual blood in non-perfused tissues [18]. According to our data, ouabain does not, in fact, cross the BBB readily; it could be a feasible alternative to digoxin in clinical applications. 

The times of the peak drug concentration (T_max_) in the blood plasma and cardiac tissue are almost identical (0.08 ± 0.01 h and 0.08 ± 0.02 h, respectively), indicating that ouabain rapidly perfuses the myocardium. Interestingly, although previous studies indicate that ouabain tends to accumulate in the cardiac tissue of dogs [18], we did not observe this in our experiment. One possible explanation for this phenomenon is different CTS retention patterns in rodents than in other mammals because of the presence of CTS in their diet. As mentioned above, the alpha1 NKA subunit of rodents, including rats and mice, is significantly less sensitive to CTS than that of other mammals. As such, we conjecture that the retention seen in dog cardiac tissue is related to the slower clearance of the alpha1-subunit-bound ouabain compared to mice. It should be noted that the time of the ouabain elimination from the plasma is significantly shorter in rats than in rabbits and dogs [21], further suggesting that lower alpha1 affinity to CTS in rodents may be implicated. It is also worth noting that we found the retention time of the ouabain in the mouse cardiac tissue to be shorter than the retention time of the 3H-digoxin in the mouse cardiac tissue, as identified in an earlier study [22]. Despite shorter retention time, ouabain has an advantage over digoxin in heart failure treatment by preventing cardiac tissue hypertrophy [4,23,24]. 

We found the T_max_ values of the ouabain in both kidney and liver tissues to be approximately 20 min, which is 15 min more than T_max_ for the plasma and cardiac tissue. They also both had higher Ft/p and C_max_ values. Thus, the distribution coefficients of ouabain were 7.17 for liver tissue and 3.0 for kidney tissue (Table 1). These data align with the assumption that both of these organs actively participate in eliminating ouabain from the body. The T_1/2_ and MRT values are slightly higher for the liver tissue (1.24 ± 0.7 h and 0.98 ± 0.33 h, respectively) than for the kidney tissue (1.32 ± 0.76 h and 1.41 ± 0.71, respectively). The higher AUC values for the liver tissue compared to the kidney tissue may indicate the predominance of the hepatic route in the elimination of the ouabain. This is supported by the apparent clearance (CL/F) value, which is the lowest for the liver tissue (584 ± 97 g/kg/h) and is almost 3 times higher for the kidney tissue (1402 ± 229 g/kg/h). We use the clearance-to-bioavailability (CL/F) ratio because the absolute value of CL, like Vd, can only be determined when using intravenous administration. The liver, being the primary elimination pathway of the ouabain from the body, is supported by previous studies on laboratory rodents, which indicate that the main route of the elimination is through the bile [25]. The decrease in ouabain’s absorption speed in the liver tissue, which we observed as the biexponential phase of the absorption in the pharmacokinetic curve (seen as a “flat top” in Figure 2D), may be due to the saturation of active transport mechanisms, as previously shown by Kupferberg H. et al., in rat liver slices [26]. 

If the main elimination pathway of the ouabain is through bile secretion, we would expect its concentration to be higher in fecal matter than in urine. Unfortunately, in this study, we did not have the means to perform an evaluation of ouabain concentrations in the urine and fecal matter of the animals, so we cannot confirm this assumption. The literature data indicate that in rats, ouabain is primarily excreted in bile with little metabolization [26]. In humans, however, ouabain appears to be primarily eliminated via the kidneys [27]. The two concentration peaks observed for the kidney tissue (Figure 2C) may be an indicator of the renal tubular recirculation of the ouabain [28]. Previously, Steiness et al. [29] have shown that in humans, digoxin excretion via the urine is in equal parts due to active tubular secretion and glomerular filtration. These results were later confirmed in rats [30]. Data based on micropuncture studies have shown that 3H-labeled digoxin in rats is absorbed from the proximal convoluted tubule but not from the loop of Henle [30]. These findings may explain the presence of two concentration peaks for the kidney tissue. In a study by Li et al. from 2021, the pharmacokinetics of the digoxin were studied in Sprague–Dawley rats, following the oral administration of the drug at a dose of 45 μg/kg. The authors showed that with respect to excretory organs, digoxin exhibits liver tropism, with a peak concentration in the liver tissue of 150 ng/g, which is over 6 times higher than that in the kidney tissue (22 ng/g) [31].

As such, we showed that ouabain does not pass through the BBB, meaning that it lacks CNS side effects when administered systemically. This should be considered when investigating the physiological effects of the ouabain, specifically on the CNS of laboratory rodents, and a direct intracerebral administration route should be used. This also implies that endogenous CTSs possessing physicochemical properties similar to those of ouabain, including polarity and structure, are unlikely to pass through the BBB in non-pathological circumstances. Also, ouabain’s tropism in cardiac tissue and lack of cardiac muscular hypertrophy induction suggests that ouabain may be preferable to digoxin in clinical applications.

## 4. Materials and Methods

### 4.1. Animals

A total of 42 C57/black male mice, 3–6 months old, were used in this study. All the animals were housed under standard vivarium conditions. The day/night cycle was 12 h, and food and water were provided ad libitum. Intraperitoneal injection was used to administer ouabain at a dosage of 1.25 mg/kg, 3 times less than the previously published LD50 for mice (3.75 mg/kg) [32]. This dose is significantly higher than therapeutic doses used for treating patients, which are 3–6 mg perorally [33,34]. After the injection, mice were sacrificed by decapitation, and the heart, kidneys, liver, brain, and blood were harvested at the following time points: 5 min, 15 min, 30 min, 1 h, 3 h, 6 h, and 24 h (*n* = 6). All the experiments were approved by the St. Petersburg State University Ethical Committee, protocol number 131-03-1 from 25 March 2019. 

### 4.2. Sample Preparation

Ouabain was extracted from biological material using deproteinization. Tissues were homogenized in 10% trichloroacetic acid to precipitate plasma proteins. D_2_–ouabain (100 ng/mL) [35] at a ratio of 1 m:2 v was added as the internal standard. The resulting suspension was ultra-centrifuged at 16,000 g to separate the denatured proteins. The supernatant was then transferred to a chromatography vial and placed in the chromatographic auto-sampler to perform further chromatography–mass spectrometry analysis. 

### 4.3. Ouabain Detection

The ouabain concentration was measured using liquid chromatography–mass spectrometry in a Shimadzu NEXERA-XR system equipped with a Shimadzu LCMS-8040 detector, a DGU-20A5r degasser, two LC-20ADxr pumps, a SIL-20AC auto-sampler, and a CTO-20AC column thermostat with two directional flow control valves (FCV-12AH and FCV 32AH, respectively). Chromatographic separation was performed in an Aeris PEPTIDE XB-C18 (50 mm × 2.1 mm × 2.6 μm) analytical column. The mobile phase consisted of two solutions: 10 mM ammonium formate acidified with formic acid to a final concentration of 0.1% (solution A) and acetonitrile–10 mM ammonium formate (90:10) (solution B) at a ratio of 90% A:10% B acidified with formic acid to a final concentration of 0.1%. The flow speed was set to 0.45 mL/min. The sample containing the ouabain was separated using the gradient method, with a progressive increase in the content of solution B from an initial value of 5% to 90% of the total mobile phase volume during the first five minutes of the analysis, followed by a two-minute equilibration of the chromatographic column at the specified concentration level and then a return to the initial value during the next three minutes of the analysis. The injection volume was 20 μL. The separation temperature was 40 °C. The duration of the chromatography was 10 min (Figure A1).

Under these conditions, both the ouabain and D_2_–ouabain (internal standard) [35] retention times constituted 3.11 ± 0.05 min. The mass-spectrometric detection of the ouabain was based on daughter ions, with *m*/*z* 439.15, 403.2, 373.2, 355.15, and 157.05, formed via the breakdown of an m/z 585.15 molecular predecessor at impact energies of 15, 16, 20, 21, and 35 eV, respectively. The second-order mass spectrum for the ouabain is presented in Figure A2.

The mass spectrometer operated in the mode of the fixation of positive ions formed by electrospray ionization (ESI). The detailed parameters of the analysis are given in Table 2. The data were processed using LabSolutions V. 5.91 software (Japan).

The internal standard method was used for the quantitative measurement of the ouabain concentrations. During the calibration, the ratio of the chromatographic peak areas of the target substance and the internal standard was measured as a function of the ouabain concentration. Linear regression, based on the least-squares method, was used for the calculations. For the quantification of the ouabain, the internal standard method was used. The calibration relationship was linear over the concentration range from 5 ng/mL to 500 ng/mL (Figure A3). The concentration of the ouabain was determined using the formula Y = 840.66 × X, where Y is the concentration of the analyte, expressed in ng/mL, and X is the area of the chromatographic peak of the ouabain normalized to the area of that of the internal standard. The relative error in the method for the determination of the ouabain concentration did not exceed 10%.

### 4.4. Pharmacokinetic Analysis and Statistical Approach 

The pharmacokinetic parameters were calculated using the noncompartmental method in the Phoenix WinNonlin 8.3 program (USA), and the statistical processing was performed in MS Office Excel. The pharmacokinetic parameters included C_max_—peak plasma concentration, T_max_—time of the peak plasma concentration, AUC—area under the concentration–time curve (trapezoidal rule), T_1/2_—the terminal elimination half-life, V—volume of the distribution, CL/F—apparent clearance, λz—terminal elimination rate constant, and mean residence time (MRT). To assess the efficiency of the drug penetration in the tissues, the tissue-to-plasma partition coefficient (Ft/p, the ratio of the AUCs between the respective peripheral site and the plasma) was used (Table A1, Table A2 and Table A3). Because in mice, the estimation of an individual pharmacokinetic profile is impossible because of the small animal size (only 1 concentration measurement is possible for 1 animal), the resampling method allowed us to obtain individual PK parameters and to estimate descriptive statistics [36].

## Figures and Tables

**Figure 1 ijms-25-04318-f001:**
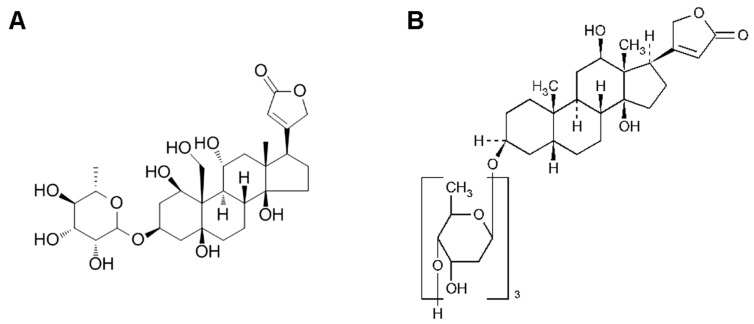
Structures of the CTS ouabain (**A**) and digoxin (**B**).

**Figure 2 ijms-25-04318-f002:**
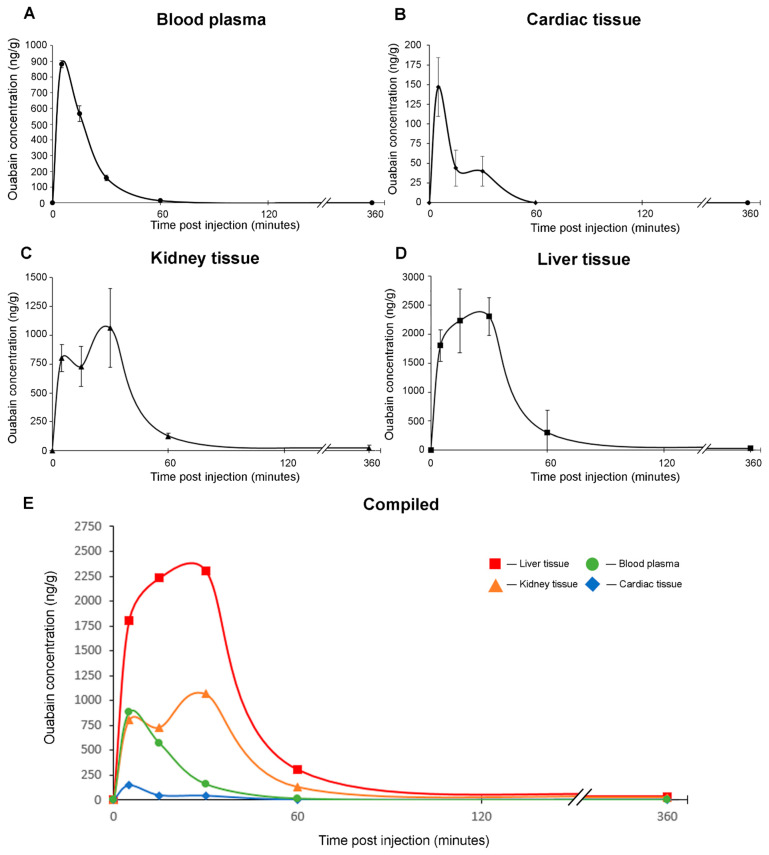
Averaged pharmacokinetic curves representing the elimination of ouabain from (**A**) blood plasma, (**B**) cardiac tissue, (**C**) kidney tissue, and (**D**) liver tissue following intraperitoneal injection of ouabain at a dose of 1.25 mg/kg bodyweight to C57/black mice. All the curves are superimposed in (**E**). *X*-axis—time from injection (min); *y*-axis—concentration (ng/g); *n* = 6. All the concentrations are presented as mean ± SD. Error bars were omitted in (**E**) to increase readability.

**Table 1 ijms-25-04318-t001:** Pharmacokinetic parameters of ouabain in tissues after intraperitoneal injection of 1.25 mg/kg bodyweight. Pharmacokinetic parameters included C_max_—peak plasma concentration, T_max_—time of peak plasma concentration, AUC—area under the concentration–time curve (trapezoidal rule), T_1/2_—the terminal elimination half-life, V—volume of distribution, CL/F—apparent clearance, λz—terminal elimination rate constant, and mean residence time (MRT). To assess the efficiency of the drug penetration in tissues, the tissue-to-plasma partition coefficient (Ft/p, the ratio of the AUCs between the respective peripheral site and the plasma) was used. Values are expressed as mean ± SD.

Tissue	C_max_, ng/g	T_max_, h	AUC_t_, ng × h/g	AUC_inf_, ngh/g	MRT, h	T_1/2_, h	Vz/F, g/kg	Cl/F, g/kg/h	λz, h^−1^	Ft/p
Brain	–	–	–	–	–	–	–	–	–	–
Blood Plasma	882.9 ± 21.8	0.08 ± 0.01	290.8 ± 14.9	294.3 ± 14.6	0.26 ± 0.01	0.15 ± 0.02	900 ± 153	4259 ± 214	4.85 ± 0.8	1.00
Cardiac	145.2 ± 44.0	0.08 ± 0.02	34.4 ± 8.7	47.6 ± 12.9	0.38 ± 0.14	0.23 ± 0.09	9463 ± 3481	29144 ± 7223	3.42 ± 1.29	0.12
Kidney	1072.3 ± 260.8	0.35 ± 0.19	872.0 ± 132.6	939.1 ± 145.4	1.41 ± 0.71	1.32 ± 0.76	2583 ± 1321	1402 ± 229	0.68 ± 0.36	3.00
Liver	2558.0 ± 382.4	0.35 ± 0.13	2084.6 ± 389.5	2207.6 ± 338.8	0.98 ± 0.33	1.24 ± 0.7	1043 ± 564	584 ± 97	0.7 ± 0.44	7.17

**Table 2 ijms-25-04318-t002:** Shimadzu 8040 triple quadrupole ESI interfacial parameters.

Parameter	Value	Unit
Nebulizer Gas Flow Rate	1.5	L/min
DL Temperature	250	C
Heat Block Temperature	400	C
Drying-Gas Flow Rate	15	L/min
CID Gas	230	kPa

## Data Availability

The original contributions presented in this study are included in the article/Appendix A; further inquiries can be directed to the corresponding author/s. The complete raw data supporting the conclusions of this article will be made available by the authors on request.

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
