# Peer review of "Evaluation of Ouabain’s Tissue Distribution in C57/Black Mice Following Intraperitoneal Injection, Using Chromatography and Mass Spectrometry"

_ijms, 2024, doi:10.3390/ijms25084318_

Round 1

Reviewer 1 Report

Comments and Suggestions for Authors

The manuscript "Evaluation of ouabain tissue distribution in C57/Black mice following intraperitoneal injection using chromatography-mass-spectrometry" by Abaimov et. al. describes a method for measuring ouabain concentrations in different organs. The manuscript is well written. However, there are some concerns.

1) The authors must provide urine and fecal analysis of ouabain and the time of retention in the body. Rodents are resistant to the effects of cardiac glycosides, but human NKA is very sensitive. Nanomolar concentrations can have adverse effects in humans. Therefore, the clearance data must be provided.

2) The authors data show very low levels of ouabain in the brains of the animals which is very encouraging. Although, the NKA alpha-1 subunit is resistant but rodents do express the highly sensitive alpha-2 and alpha-3 which can be inhibited at small concentrations. Therefore, before making a conclusion, it is advisable to perform behavior experiments such as Barnes Maze assay for behavior analysis.

Comments on the Quality of English Language

None

Author Response

The manuscript "Evaluation of ouabain tissue distribution in C57/Black mice following intraperitoneal injection using chromatography-mass-spectrometry" by Abaimov et. al. describes a method for measuring ouabain concentrations in different organs. The manuscript is well written. However, there are some concerns.

1) The authors must provide urine and fecal analysis of ouabain and the time of retention in the body. Rodents are resistant to the effects of cardiac glycosides, but human NKA is very sensitive. Nanomolar concentrations can have adverse effects in humans. Therefore, the clearance data must be provided.

We are grateful to the Reviewer for their suggestion that adding urine and fecal analysis of ouabain clearance would improve our manuscript. However, we believe this to be outside of the scope of our study, since we focused on answering the question of whether it is capable of passing through the blood-brain barrier. According to literature data, in rats ouabain is primarily excreted in bile with little metabolization (doi:10.1152/ajplegacy.1968.214.5.1048). In humans, however, ouabain appears to be primarily eliminated via the kidneys (Osol A. Remington’s Pharmaceutical Sciences: Ed. 16. Mack Publishing Company; 1980) Ouabain clearance pathways from the body warrant separate investigation, especially considering that direct intestinal excretion has been shown for CTS such as digoxin (10.1111/j.1476-5381.1996.tb15775.x). We agree that this is an important point, which is addressed in our text in the discussion section.

2) The authors data show very low levels of ouabain in the brains of the animals which is very encouraging. Although, the NKA alpha-1 subunit is resistant but rodents do express the highly sensitive alpha-2 and alpha-3 which can be inhibited at small concentrations. Therefore, before making a conclusion, it is advisable to perform behavior experiments such as Barnes Maze assay for behavior analysis.

The authors thank the Reviewer for their comment, but would like to draw their attention to the fact that we did not detect any ouabain at all in brain tissue. We agree with the Reviewer that alpha-2 and alpha-3 NKA subunits are sensitive in rodents. Effects on behavior in mice become visible when ouabain reaches nanomolar concentrations in the brain (unpublished data acquired using the same detection protocol as in present manuscript) following intracerebroventricular administration of 50 mkM ouabain (doi:10.1038/s41598-019-52058-z), which implies that if ouabain was present in concentrations which could affect behavior we would be able to register its presence in brain tissue. At the same time, we believe that behavioral tests would be inconclusive, since while the chosen dose did not cause visible alterations in animal well-being, it could still cause changes in behavior mediated by overall animal condition rather than specifically changes in the brain.

Reviewer 2 Report

Comments and Suggestions for Authors

The manuscript „Evaluation of Ouabain Tissue Distribution in C57/Black Mice 2 Following Intraperitoneal Injection Using Chromatog-3 raphy-Mass-Spectrometry“ by Denis A. Abaimov, Rogneda B. Kazanskaya, Maxim S. Nesterov, Yulia A. Timoshina, Angelina I. Platova, Irina J. Aristova, Irina S. Vinogradskaia, Tatiana N. Fedorova, Anna B. Volnova, Raul R. Gainetdinov, Alexander V. Lopachev presents new findings showing pharmacokinetics of ouabain in heart, liver, kidney and blood plasma; in brain tissue it was undetectable.

This referee has the following comments:

  1. Introduction: Check spelling in l. 58.
  2. Introduction: The knowledge gap addressed is missing as well as a specific hypothesis (if applicable), please provide.  
  3. Results: How is the tested dose of ouabain related to clinically relevant dosages?
  4. Results: Add unit to mean retention time, l.114.
  5. Results, Fig. 2, legend: Reference to panel F probably means panel E.
  6. Results, Table 1, legend: Explanations for the parameters presented are missing so that the table can be understood without reference to the text, please add.
  7. Methods: The method how mice were sacrificed should be described.

Author Response

  1. Introduction: Check spelling in l. 58.

We are grateful to the Reviewer for pointing out this mistake, and have corrected the formatting of CH2OH to CH2OH.

  1. Introduction: The knowledge gap addressed is missing as well as a specific hypothesis (if applicable), please provide.  

Thank you for the question. The specific hypothesis for this study was that ouabain would not pass through the blood-brain barrier due to increased polarity relative to other CTS, and that the results previously acquired by other investigators were false positives (as discussed in the discussion section). While there is no clear “knowledge gap”, since the pharmacokinetics of ouabain have been studied previously, new methods allow us to evaluate the tissue distribution of ouabain with fewer limitations. We introduced a clarification of the hypothesis into the second to last paragraph in the introduction section.

  1. Results: How is the tested dose of ouabain related to clinically relevant dosages?

The authors thank the Reviewer for bringing up this question. The dose tested in this study was not chosen based on clinically relevant dosages, but based on the lethal dose of ouabain to mice –1.25 mg/kg ouabain did not cause immediate, visible outward changes in behavior or well-being of the animals, while higher doses caused some mice to exhibit symptoms of poisoning (labored breathing, twitching, collapse). The doses administered to humans are significantly lower (0.25 mg i.v., 6 mg orally, DOI: 10.1055/s-0028-1129021, 3 mg orally DOI:10.58372/2835-6276.1069), since the human alpha1 and alpha2 NKA subunits are sensitive to CTS.

  1. Results: Add unit to mean retention time, l.114.

We have added the “h” (hour) unit to the mean retention time value.

  1. Results, Fig. 2, legend: Reference to panel F probably means panel E.

We are grateful to the Reviewer for noticing this error and have corrected it to E.

  1. Results, Table 1, legend: Explanations for the parameters presented are missing so that the table can be understood without reference to the text, please add.

We added the description of pharmacokinetic parameters provided in the methods section to the legend of Table 1.

  1. Methods: The method how mice were sacrificed should be described

We added the clarification that mice were decapitated in the methods section.

Round 2

Reviewer 1 Report

Comments and Suggestions for Authors

Answers provided are not satisfactory and the conclusion that "ouabain is better than digoxin in part due to its inability to cross the BBB" is not supported by the data.